# Concentrations of Ciprofloxacin in the World’s Rivers Are Associated with the Prevalence of Fluoroquinolone Resistance in *Escherichia coli*: A Global Ecological Analysis

**DOI:** 10.3390/antibiotics11030417

**Published:** 2022-03-20

**Authors:** Chris Kenyon

**Affiliations:** 1HIV/STI Unit, Institute of Tropical Medicine, Antwerp 2000, Belgium; ckenyon@itg.be; Tel.: +32-3-248-0796; Fax: +32-3-248-0831; 2Division of Infectious Diseases and HIV Medicine, University of Cape Town, Anzio Road, Observatory, Cape Town 7700, South Africa

**Keywords:** rivers, one health, *E. coli*, fluoroquinolones, antimicrobial resistance, AMR

## Abstract

Background: Extremely low concentrations of ciprofloxacin may select for antimicrobial resistance. A recent global survey found that ciprofloxacin concentrations exceeded safe levels at 64 sites. In this study, I assessed if national median ciprofloxacin concentrations in rivers were associated with fluoroquinolone resistance in *Escherichia coli*. Methods: Spearman’s regression was used to assess the country-level association between the national prevalence of fluoroquinolone resistance in *E. coli* and the median ciprofloxacin concentration in the country’s rivers. Results: The prevalence of fluoroquinolone resistance in *E. coli* was positively correlated with the concentration of ciprofloxacin in rivers (ρ = 0.36; *p* = 0.011; *n* = 48). Discussion: Steps to reducing the concentrations of fluoroquinolones in rivers may help prevent the emergence of resistance in *E. coli* and other bacterial species.

## 1. Background

The reasons underpinning the global variations of fluoroquinolone resistance in Gram-negative bacteria are incompletely understood [1,2,3]. Whilst human consumption of quinolones plays an important role, much of the variation remains unexplained, with recent analyses suggesting that various socio-economic factors play a critical role [1,4,5]. For example, studies have found that better infrastructure, such as improved sanitation and potable water, were significantly associated with lower antimicrobial resistance (AMR) indices [1,5].

A relatively unexplored pathway towards AMR in Gram-negatives is antimicrobial exposure in ambient water bodies [6,7]. A number of studies have found that extremely low concentrations of antimicrobials can select for de novo AMR and enrichment of resistant versus susceptible strains in a range of Gram-negative bacteria, including *Escherichia coli* [8,9,10]. The lowest concentration of an antimicrobial that selects for resistant over susceptible strains in competition assays is referred to as the minimum selection concentration (MSC) [8,10,11]. In the case of *E. coli*, the ciprofloxacin MSC is 230-fold lower than the minimum inhibitory concentration (MIC) [8].

The appreciation that the MSC is considerably lower than the MIC has led experts to recommend that the maximum concentration of antimicrobials allowed in water should be approximately 10- to 100-fold lower than the lowest minimum inhibitory concentrations (MICs) of bacteria present in large datasets such as that of EUCAST [6,9]. 

A number of studies have found that high concentrations of antimicrobials in ambient water are associated with AMR in various bacterial species in the environment [7,12,13]. Little is known as to whether or not antimicrobials in ambient water can cause AMR in bacteria colonizing humans. One study has found that surfers have a higher carriage of CTX-M-producing *E. coli* than non-surfers, and that the likely exposure was the ingestion of water whilst surfing [14]. *E. coli* is a particularly relevant species to test because of its fecal–oral transmission and the fact that it is one of the most common bacterial causes of diarrhea worldwide [15].

In this study, I assessed if there was a country-level association between the prevalence of fluoroquinolone resistance in *E. coli* and the concentration of ciprofloxacin in river water. To accomplish this, I made use of a recently published global survey of pharmaceuticals in the world’s rivers that found high levels of ciprofloxacin at numerous sites [6]. This survey used a standard collection and assay to measure the concentrations of 61 active pharmaceutical ingredients at over 1000 locations on 258 of the world’s rivers in 104 countries [6]. This included the quinolone antimicrobial ciprofloxacin. One of the most striking findings was that the concentration of ciprofloxacin exceeded the proposed safe limit at 64 sites of all the sites sampled. I used Spearman’s correlation to compare concentrations of ciprofloxacin with the prevalence of *E. coli* resistance to ciprofloxacin from a global database. 

## 2. Methods

### 2.1. Antimicrobial Resistance Data

The prevalence of *Escherichia coli* fluoroquinolone resistance per country was obtained from the Center for Disease Dynamics, Economics & Policy’s (CDDEP) ResistanceMap database.Available online: https://resistancemap.cddep.org/AntibioticResistance.php (accessed on 10 February 2022).

CDDEP obtains and aggregates data on the prevalence of antibiotic resistance from several sources. The data are then harmonized to present similar definitions of resistance across countries and regions to enable comparisons between countries. As an example, the AMR data for European countries are from European Antimicrobial Resistance Surveillance Network (EARS-Net). These data are collected by each country and only include *E. coli* invasive isolates from blood and CSF. EARS-Net encourages the use of the European Committee on Antimicrobial Susceptibility Testing (EUCAST) breakpoints for the calculation of the proportion of isolates with AMR. For additional details as to the methodology and definitions used to define antimicrobial resistance [16]. The CDDEP data provide a resistance prevalence estimate from a single year for each country, which is typically the year 2017.

### 2.2. Quinolone Concentrations in Rivers

Wilkinson et al. used a standard collection and assay to measure the concentration of a number of pharmaceutical agents, including ciprofloxacin, at 1052 locations on 258 of the world’s rivers in 104 countries [6]. In brief, between February 2018 and January 2020, they mailed a standard collection kit to collaborators in 104 countries and asked them to follow a standard sample-collection protocol. The sampling sites typically included sites upstream, within, and downstream from a populated area. The samples were kept frozen after collection until they were sent frozen via air shipment to a single analytical center in the United Kingdom for analysis using a single analytical method (high-pressure liquid chromatography–tandem mass spectrometry). I used their data to calculate the median ciprofloxacin concentration per country. The assay used had a limit of detection of 10.1 ng/L for ciprofloxacin. A value of 0 ng/L was used for all sites, where the concentration of ciprofloxacin was measured to be below the limit of detection. 

### 2.3. Statistical Analyses

Spearman’s regression was used to assess the country-level association between the national prevalence of fluoroquinolone resistance in *E. coli* and the median ciprofloxacin concentration in the countries’ rivers. Statistical analyses were performed in Stata 16.0, and a *p*-value of <0.05 was regarded as statistically significant. 

## 3. Results

Ciprofloxacin concentration and resistance data were available from 104 and 58 countries, respectively. Forty-eight countries had data available for both variables. In these 48 countries, the prevalence of both fluoroquinolone resistance (median, 29%; IQR, 20.5–45.5%) and ciprofloxacin concentrations in rivers (median, 0 ng/L; IQR, 0–45.3 ng/L) varied considerably between countries (Table 1 and Figure 1). 

### Spearman’s Correlations

The prevalence of fluoroquinolone resistance in *E. coli* was positively correlated with the concentration of ciprofloxacin in rivers (ρ = 0.36; *p* = 0.011; *n* = 48).

## 4. Discussion

In this global ecological study, I found a positive correlation between the concentration of ciprofloxacin in countries’ rivers and the prevalence of fluoroquinolone resistance in *E. coli*. A number of pathways could explain this association. Countries with high ciprofloxacin concentrations in their rivers may consume more ciprofloxacin, and this higher consumption may result in both AMR in *E. coli* in humans and higher concentrations of ciprofloxacin excreted and hence detected in rivers [7]. Similarly, these countries may use more fluoroquinolones for animal husbandry, and this may generate AMR in *E. coli* in animals that can be transmitted to humans as well as be excreted into rivers [17,18]. Studies have found positive associations between the prevalence of fluoroquinolone resistance in *E. coli* in humans and *E. coli* from poultry and pigs [19]. A systematic review performed on this topic found that fluoroquinolone resistance could be transferred from *E. coli* in food-producing animals to humans [20]. Finally, the low levels of ciprofloxacin in rivers might generate AMR in *E. coli*, which is then ingested by humans. Our ecological study is unable to disentangle which of these pathways is predominant.

Further limitations of our study include the fact that I did not have longitudinal, individual-level, or experimental data, the fact that the exposure variable postdates the outcome variable, the small number of countries with resistance data, and the relatively low number and non-representativity of sites per country used to establish ciprofloxacin concentrations. The limit of detection for ciprofloxacin was fairly high, which may have introduced a bias by assuming that all sites with undetectable levels of ciprofloxacin had a concentration of 0 ng/L. The fluoroquinolone resistance prevalence estimates from CDDEP are based on a number of different methodologies, making comparisons between countries sub-optimal. I was not able to adjust my analyses for differences in susceptibility-testing strategies or breakpoints between countries, as CDDEP does not provide this information.

Finally, I did not control for a number of potential confounders in the relationship between ciprofloxacin concentration and resistance, such as the inadequate processing of sewage, poor sanitation, inadequate regulation of antimicrobials, consumption of other classes of antimicrobials, and variations in environmental temperatures [1,2,5,21,22,23,24]. 

In conclusion, I found that the prevalence of fluoroquinolone resistance in *E. coli* was positively correlated with the concentration of ciprofloxacin in rivers. Despite the limitations listed above, this ecological study provides motivation for additional research to define safe upper limits for antimicrobials in rivers and ground water. This research should include in vitro MSC-determination assays that exceed the 3 to 7 days currently assessed and include stochastic variations in antimicrobial exposure [7,11]. Finally, they should include assessments of the recently established minimum increased persistence concentration (MIPC), which might result in the enrichment of resistance well below MSC concentrations [11].

## Figures and Tables

**Figure 1 antibiotics-11-00417-f001:**
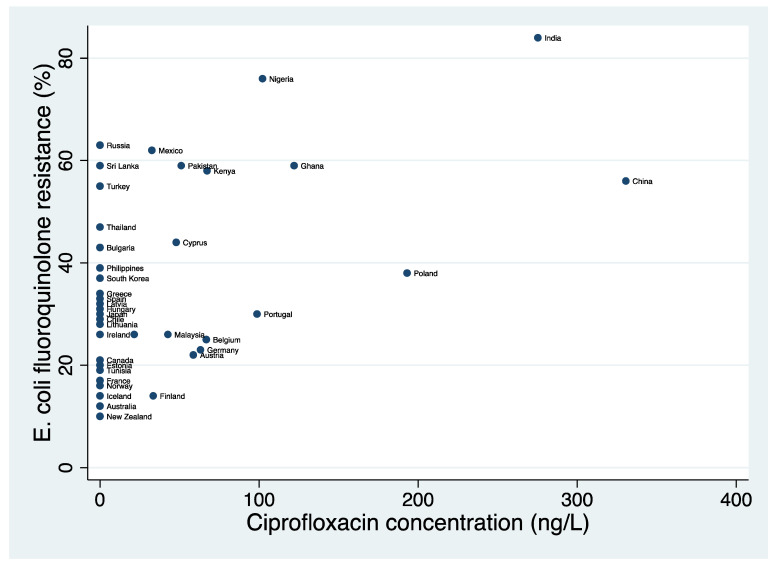
Scatter plot of national median concentration of ciprofloxacin in rivers (ng/L) versus prevalence of *Escherichia coli* fluoroquinolone resistance (%).

**Table 1 antibiotics-11-00417-t001:** Country-level concentration of ciprofloxacin (ng/L) in rivers and percent of *Escherichia coli* isolates obtained from human infections resistant to fluoroquinolones.

Country	Year ^a^	*n* Sites Tested ^b^	Ciprofloxacin Concentration (Median, ng/L)	Ciprofloxacin Concentration (Mean, ng/L)	*E. coli* Fluoroquinolone Resistance (%)
Australia	2017	0	0	0	12
Austria	2017	2	58.7	58.7	22
Belgium	2017	2	66.8	66.8	25
Bulgaria	2017	0	0	0	43
Canada	2014	0	0	0	21
Chile	2014	0	0	0	29
China	2017	4	330.5	329.5	56
Croatia	2017	0	0	0	29
Cyprus	2018	3	47.9	59.8	44
Czech Republic	2017	7	21.5	22.8	26
Denmark	2017	0	0	0	14
Estonia	2017	0	0	0	20
Finland	2017	4	33.5	41	14
France	2017	0	0	0	17
Germany	2017	4	63.2	96.9	23
Ghana	2016	1	122	122	59
Greece	2017	0	0	0	34
Hungary	2017	0	0	0	31
Iceland	2017	0	0	0	14
India	2017	6	275.2	303.3	84
Ireland	2017	0	0	0	26
Italy	2017	0	0	0	47
Japan	2017	0	0	0	30
Kenya	2015	7	67.3	78.2	58
Latvia	2017	0	0	0	32
Lithuania	2017	0	0	0	28
Malaysia	2017	10	42.7	52.8	26
Mexico	2015	3	32.6	26.7	62
Netherlands	2017	0	0	0	16
New Zealand	2015	0	0	0	10
Nigeria	2017	8	102.2	144.8	76
Norway	2017	0	0	0	16
Pakistan	2017	4	51.1	52.0	59
Philippines	2017	0	0	0	39
Poland	2017	1	193	193	38
Portugal	2017	4	98.7	99.0	30
Romania	2017	0	0	0	28
Russia	2017	0	0	0	63
Slovenia	2017	0	0	0	26
South Africa	2016	0	0	0	28
South Korea	2017	0	0	0	37
Spain	2017	0	0	0	33
Sri Lanka	2009	0	0	0	59
Sweden	2017	0	0	0	17
Switzerland	2017	0	0	0	19
Thailand	2017	0	0	0	47
Tunisia	2017	0	0	0	19
Turkey	2016	0	0	0	55

^a^ Year for *E. coli* resistance prevalence study; ^b^ number of sites tested to establish ciprofloxacin concentration in rivers.

## Data Availability

The data used are publicly available from: https://resistancemap.cddep.org/AntibioticResistance.php (accessed on 10 February 2022).

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
