# Peer review of "Concentrations of Ciprofloxacin in the World’s Rivers Are Associated with the Prevalence of Fluoroquinolone Resistance in Escherichia coli: A Global Ecological Analysis"

_antibiotics, 2022, doi:10.3390/antibiotics11030417_

Round 1
Reviewer 1 Report
General Comments
The author describes interesting study on relationship of fluoroquinolones in rivers and fluoroquinolone resistance in E. coli. The authors state the limitation of study; still the data give an interesting insight in the research topic
The paper is well written and structured and adequately presented. There are several writing errors that should be checked, see details below.
Additional Comments
Abstract
- Check for writing errors (missing space)
Introduction
- Lines 46-48. Check grammar in the sentence
- Lines 68-70 Writing error
Results
- Lines 98-100 Writing error
Discussion
- Lines 126-128 Writing error
- Generally it is short but adequate.
Additional questions:
- Since we are talking about global problem, is there any information about the source of isolated coli that were tested for fluoroquinolone resistance? Near the rivers that were tested, or?
- Another question is about bacterial strain. coli is, so to say, a large group of different microorganism. How pathogenic were the tested strains. Short info should be given in this study.
Reviewer 2 Report
This paper addresses an important issue that has an impact on health. Amr in aquatic ecosystems represents a means for the spread of resistance.
- It is important that the author include a brief conclusion.
- The sampling of river water was from 2018 to 2020 and the prevalence date of e. coli resistant to cddep quinolones was 2017, how can they be compared, if they are different years?
- Please clarify 64 sampling sites? Lines 11 and 59
- table 1 repeats the ciprofloxacin concentration column (mean ng/ l) with very similar data, why? It is not clear from the results the reason for these data. Please clarify
